# Active Hierarchical Exploration with Stable Subgoal Representation Learning

**Siyuan Li†, Jin Zhang†, Jianhao Wang†, Yang Yu‡, Chongjie Zhang†**
†Tsinghua University, ‡Nanjing University
{sy-li17,jin-zhan20,wjh19}@mails.tsinghua.edu.cn
{yuy}@lamda.nju.edu.cn, chongjie@tsinghua.edu.cn

## Abstract

Goal-conditioned hierarchical reinforcement learning (GCHRL) provides a promising approach to solving long-horizon tasks. Recently, its success has been extended to more general settings by concurrently learning hierarchical policies and subgoal representations. Although GCHRL possesses superior exploration ability by decomposing tasks via subgoals, existing GCHRL methods struggle in temporally extended tasks with sparse external rewards, since the high-level policy learning relies on external rewards. As the high-level policy selects subgoals in an online learned representation space, the dynamic change of the subgoal space severely hinders effective high-level exploration. In this paper, we propose a novel regularization that contributes to both stable and efficient subgoal representation learning. Building upon the stable representation, we design measures of novelty and potential for subgoals, and develop an active hierarchical exploration strategy that seeks out new promising subgoals and states without intrinsic rewards. Experimental results show that our approach significantly outperforms state-of-the-art baselines in continuous control tasks with sparse rewards.

## 1 Introduction

Goal-conditioned hierarchical reinforcement learning (GCHRL) has long demonstrated great potential to solve temporally extended tasks (Dayan & Hinton, 1993; Schmidhuber & Wahnsiedler, 1993; Vezhnevets et al., 2017; Péré et al., 2018; Nair & Finn, 2019), where a high-level policy periodically sets subgoals to a low-level policy, and the low-level policy is intrinsically rewarded for reaching those subgoals. Early GCHRL studies used a hand-designed subgoal space, such as positions of robots (Nachum et al., 2018; Levy et al., 2019) or objects in images (Kulkarni et al., 2016a). To alleviate the dependency on domain-specific knowledge, recent works investigate learning subgoal representations along with hierarchical policies (Nachum et al., 2019a; Dilokthanakul et al., 2019; Li et al., 2021), which have shown promise in a more general setting. Although the exploration ability of HRL is boosted via decomposing tasks and reusing low-level policies, existing GCHRL methods struggle in long-horizon tasks with sparse external rewards. The high-level policy learning in GCHRL relies on external rewards, and those tasks are hard to solve with naive high-level exploration strategies. Furthermore, as the high-level policy makes decisions in a latent subgoal space learned simultaneously with the hierarchical policy, the instability of the online-learned subgoal space results in severe non-stationarity of the high-level state transition function and hinders effective high-level exploration.

To address these challenges, we develop an active Hierarchical Exploration approach with Stable Subgoal representation learning (HESS). Our approach adopts a contrastive objective for subgoal representation learning inspired by Li et al. (2021). To learn a subgoal space that could effectively guide hierarchical exploration, we propose a novel state-specific regularization that contributes to both stable and efficient subgoal representation learning. The proposed regularization constrains the changes for those embeddings that have already well satisfied the representation learning objective. As the hierarchical agent gradually expands its exploration to new state regions, the regularization allows for updates of the embeddings that underfit the learning objective. Benefiting from the proposed regularization, we could improve the representation learning efficiency without hurting its

stability via the prioritized sampling technique (Hinton, 2007), which prioritizes training samples with larger losses.

Building upon our stable subgoal representation learning, we design an active exploration strategy for high-level policy learning. As shown in previous work (Strehl & Littman, 2008; Bellemare et al., 2016; Ostrovski et al., 2017), the visit count provides a simple and effective novelty measure for exploration. However, with online subgoal representation learning, the visit count for subgoals is defined in a changing space. Therefore, such a novelty measure alone is not sufficient for efficient high-level exploration, although the stability regularization could improve its accuracy. Our insight is that desirable novel subgoals should be reachable and effectively guide the agent to unexplored areas. Thus we design a novel *potential* measure to regularize the novelty measure, which indicates the reachability to the neighbor regions of the visited state embeddings. With the regularized novelty measure, our proposed active exploration strategy directly selects state embeddings with high potential and novelty as subgoals without introducing intrinsic rewards. It is more efficient than the *reactive* exploration methods that need to learn how to maximize the intrinsic rewards before performing exploratory behaviors (Tang et al., 2017; Pathak et al., 2017; Burda et al., 2018), since in the active exploration approach, the subgoal measures have direct effects on the behavior policy. Furthermore, the active exploration strategy naturally avoid introducing additional non-stationarity (i.e., induced by dynamically changing intrinsic rewards) into HRL.

We compare the proposed method HESS with state-of-the-art baselines in a number of difficult control tasks with sparse rewards. Note that the environment setting in this paper is much more challenging to exploration than the multi-task and deceptive dense-reward ones in the baselines. Experimental results demonstrate that HESS significantly outperforms existing baselines. In addition, we perform multiple ablations illustrating the importance of the various components of HESS.

## 2 BACKGROUND

We consider a Markov Decision Process (MDP) defined as a tuple $(S, A, P, r, \gamma)$, where $S$ is a state space, $A$ is an action space, $P(s'|s, a)$ is an unknown transition function, $r : S \times A \to \mathbb{R}$ is a reward function, and $\gamma \in [0, 1)$ is a discount factor. Let $\pi(a|s)$ denote a stochastic policy over actions given states. The objective of reinforcement learning (RL) is to learn a policy that maximizes the expected cumulative discounted rewards: $\max_\pi \mathbb{E}_{P,\pi}[\sum_{t=0}^{T} \gamma^t r(s_t, a_t)]$, where $T$ denotes the horizon length.

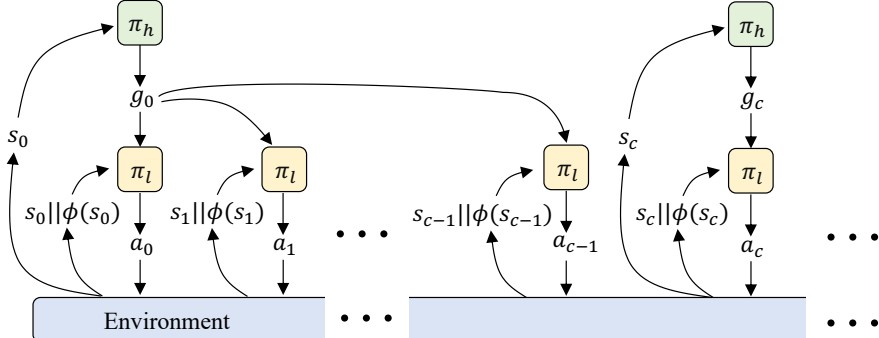

Figure 1: The hierarchical framework of GCHRL.

In GCHRL, a two-level hierarchical policy $\pi_{hier}$ is composed of a high-level policy $\pi_h(g_t|s_t)$ and a low-level policy $\pi_l(a_t|s_t^l, g_t)$, as illustrated in Figure 1. A latent subgoal space $G$ is abstracted by a representation function $\phi(s) : S \to \mathbb{R}^d$. The high-level policy $\pi_h$ samples a subgoal $g_t$ from a neighborhood $\Delta(\phi(s_t))$ of the current latent state $\phi(s_t)$ every $c$ steps, i.e., when $t \equiv 0 \pmod c$: $\Delta(\phi(s_t)) = \{g_t \in G | D(g_t, \phi(s_t)) \le r_g\}$, where $c$ is a fixed constant, $D$ is a distance function, and $r_g$ is a radius of the neighborhood. The sampled subgoal is repeated, $g_t = g_{t-1}$, when $t \not\equiv 0 \pmod c$. $\pi_h$ is trained to optimize the expected extrinsic rewards. The low-level controller takes a primitive action $a_t \in A$ every step, and is intrinsically rewarded to reach subgoals set by the high level, $r_t^l = -D(g_t, \phi(s_t))$. To provide dense non-zero rewards for low-level policy learning, we employ L2 distance as D. Furthermore, to keep the low-level reward function $r^l$ stationary while learning $\phi$, we concatenate states and state embeddings together as the low-level states, $s^l = s||\phi(s)$. Previous work (Li et al., 2021) minimizes a contrastive triplet loss $L_{tri}$ to learn $\phi(s)$, where positive

pairs are adjacent states in trajectories, and negative pairs are states $c$ steps apart.

$$L_{tri}(s_t, s_{t+1}, s_{t+c}) = ||\phi(s_t) - \phi(s_{t+1})||_2 + max(0, \delta - ||\phi(s_t) - \phi(s_{t+c})||_2), \quad (1)$$

where $\delta$ is a margin parameter. In this work, we also adopt the contrastive loss in Equation 1.

## 3 METHOD

In GCHRL, the low-level policy is optimized with intrinsic rewards generated by subgoals, but external rewards for the high level are still sparse and hard to explore. Furthermore, the changing subgoal space makes it even more challenging for high-level exploration. In this section, we first present a novel regularization that contributes to stable and efficient subgoal representation learning, and then introduce two measures for subgoals, novelty and potential. Finally, we design an active hierarchical exploration strategy based on these two measures.

### 3.1 STABLE SUBGOAL REPRESENTATION LEARNING

Both stability and efficiency are crucial to subgoal representation learning. The stability of subgoal representation contributes to the stationarity of both the high-level transition function and the low-level reward function. Meanwhile, a fast learned subgoal representation could provide effective guidance to exploration. However, stability seems at odds with efficiency, e.g., training neural networks with smaller learning rates leads to better stability but is slower (Goodfellow et al., 2016). In this subsection, we develop a novel state-specific regularization that resolves the stability-efficiency dilemma in subgoal representation learning.

To achieve stable representation learning, we propose a regularization $L_s$ which restricts the representation change during each update. $L_s$ works by anchoring the current representation $\phi(s)$ to the old subgoal representation $\phi_{old}(s)$ before the update as follows:

$$L_s = \mathbb{E}_{s \sim \mathcal{B}}[\lambda(s)||\phi(s) - \phi_{old}(s)||_2], \quad (2)$$

where $\mathcal{B}$ is a replay buffer, and $\lambda(s)$ is a function that controls the weight of regularization for different states. States with smaller representation losses fit the learning objective well, so $\lambda(s)$ should be larger for those states. In practice, before each representation update, we rank the triplets in the buffer with $L_{tri}(s_t, s_{t+1}, s_{t+c})$. For the top $k\%$ of the triplets with the minimum representation losses, we set $\lambda(s) = \lambda_0 > 0$ for the anchor states $(s_t)$ in these triplets, and for the other states, $\lambda(s) = 0$.

The stability regularization enables us to use prioritized sampling (Hinton, 2007) to improve representation learning efficiency without hurting its stability. The overall loss for subgoal representation learning is:

$$L_\phi = \mathbb{E}_{(s_t, s_{t+1}, s_{t+c}) \sim \mathcal{B}_p}[L_{tri}(s_t, s_{t+1}, s_{t+c})] + L_s, \quad (3)$$

where $\mathcal{B}_p$ is a prioritized replay buffer, and states with larger $L_{tri}$ have higher probabilities of being sampled for training.

### 3.2 MEASURES FOR SUBGOALS

**Novelty Measure:** Inspired by count-based exploration methods (Strehl & Littman, 2008; Bellemare et al., 2016; Ostrovski et al., 2017; Tang et al., 2017), we formulate the novelty of subgoals with visit counts. As desirable subgoals should incentivize the agent to explore faraway novel states, we consider both immediate counts $n(\phi(s_i))$ and expected cumulative counts of future states, and the novelty measure $N(\phi(s_i))$ is defined as follows:

$$N(\phi(s_i)) = \mathbb{E}_{\pi_{hier}}[\sum_{j=0}^{\lfloor (T-i)/c \rfloor} \gamma^j n(\phi(s_{i+jc}))]. \quad (4)$$

In practice, we partition the low-dimensional continuous latent space into discrete cells, i.e., the state embeddings are mapped into cells containing them. By maintaining how many times each cell is visited, we could estimate the visit count $n(\phi(s))$. In practice, the novelty measure is approximated with the data from the replay buffer, and the implementation detail is described in Appendix B.

**Potential Measure:** With online representation learning, the novelty measure is a mixture of counts in the past and current representation spaces, so it might mislead the exploration, as demonstrated in Figure 3. Our insight is that desirable novel subgoals should be reachable and effectively guide the agent to unexplored areas. Therefore, we design a *potential* measure for subgoals to regularize the novelty measure. In the following, we first introduce a subgoal generation mechanism, which is involved in the definition of the potential measure.

To guide the low-level controller to reach unexplored states, the subgoals pursued by the low level had better be in unexplored areas as well. Therefore, we propose to add some perturbations to the subgoal $g_t$ selected from the replay buffer and obtain an imagined subgoal $g_e$, and then pass $g_e$ to the low-level policy. To enable $g_e$ in an unexplored or less explored area, the perturbation is conducted as extending $g_t$ in the direction of $g_t - \phi(s_t)$, as illustrated in Figure 2.

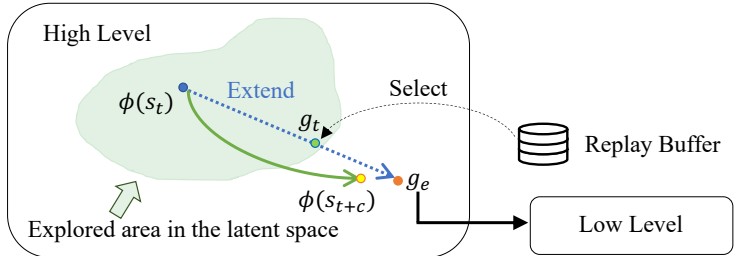

Figure 2: A schematic illustration of the subgoal selection and perturbation.

The imagined subgoal is $g_e = g_t + d_e(g_t - \phi(s_t))/\|g_t - \phi(s_t)\|_2$, where $d_e$ denotes an extended distance. As $g_e$ is imagined, and may have not been visited before, it could be inherently unreachable due to the transition function of the MDP or online representation learning, e.g., there may be obstacles in navigation tasks. To encourage the agent to explore in promising and reachable directions, we define a measure of potential $U(g_t)$ for the selected subgoal $g_t$ as the expected negative distance between the ending state $\phi(s_{t+c})$ and $g_e$:

$$U(g_t) = \mathbb{E}_{s_t, a_t, \ldots, s_{t+c}}[-D(\phi(s_{t+c}), g_e)], \text{ where}$$
$$s_t \sim \rho_h, a_t \sim \pi_l(a_t | s_t^l, g_e), s_{t+1} \sim P(s_{t+1} | s_t, a_t). \quad (5)$$

$\rho_h$ is the high-level state distribution under the hierarchical policy $\pi_{hier}$. Building on (Li et al., 2021), the representation learned with the contrastive objective preserves temporal coherence, and the distances between nearby features approximately represent the number of transitions between them (Oord et al., 2018). Hence, with higher potential $U(g_t)$, $g_e$ is more reachable, and thus exploring the direction of $g_t$ is more promising to expand the explored areas. The potential $U(g_t)$ in Equation 5 is estimated from the data in buffer as well. To calculate the novelty, we partition the continuous representation space into discrete cells. Similarly, we maintain the potential of a cell by averaging the potential of features in that cell, and use the potential of the cell to represent that of the states inside it.

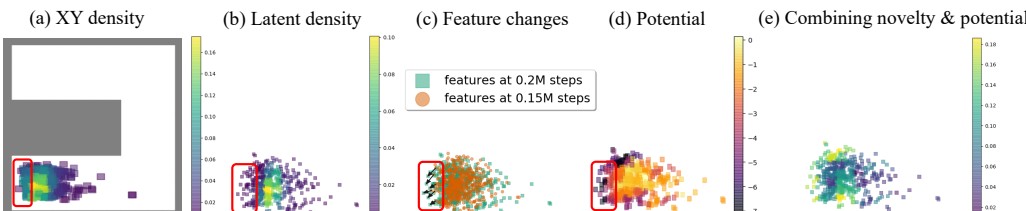

Figure 3: Visualization of visitation density and potential in the Ant Maze task. (a) Visitation density in the $x, y$ coordinate space of the Ant robot. (b) Visitation density in the subgoal representation space. (c) Feature changes between 0.15M and 0.2M steps for the same batch of states. (d) Potential for the sampled state embeddings. (e) Combination of novelty and potential in Section 3.3. Our method selects state embeddings with darker colors as subgoals.

**Illustrative example:** In Figure 3(a) and 3(b), we visualize the visitation density at 0.2M steps in the Ant Maze task in Figure 5. The visitation density is the counts normalized by the total number of transitions in the buffer. Comparing Figure 3(a) to 3(b), we found a mismatch between the density in the $x, y$ coordinate space and the representation space, especially in the red box, since the updated

representation at 0.2M steps has projected the state embeddings to somewhere new, and the feature changes are noted by the black arrows in Figure 3(c). Furthermore, the counts in frontiers of the latent explored areas would not increase with more exploration, as the online learned representation keeps changing. With the potential measure, we could distinguish the promising novel areas from the unpromising ones, illustrated in Figure 3(d).

### 3.3 ACTIVE EXPLORATION STRATEGY

In the *reactive* exploration methods with intrinsic rewards, the agent needs to learn how to maximize cumulative intrinsic rewards before performing exploratory behaviors. Therefore, the effects of intrinsic rewards on behavior policy is indirect. Furthermore, the changing intrinsic rewards would introduce additional non-stationarity into HRL. To make the proposed measures directly influence the behavior policy and avoid the non-stationary issue, we develop an active exploration strategy without intrinsic rewards for high-level policy learning. Based on the two measures in Section 3.2, the proposed exploration strategy selects and explores a novel subgoal $g_t$ with high potential, specified as follows:

$$g_t = \arg\min_{\phi(s)} \widetilde{N}(\phi(s)) - \alpha U(\phi(s))$$

$$\text{subject to} \begin{cases} D(\phi(s), \phi(s_t)) \leq r_g \\ s \in \mathcal{B}, \end{cases} \tag{6}$$

where $s_t$ is the current state and $\alpha$ is a scaling factor. To balance these two measures more easily, we normalize future count $N(\phi(s))$ by the total number of transitions in the buffer, and denote it with $\widetilde{N}(\phi(s))$. We visualize the proposed exploration incentive in Figure 3(e). Taking the state embeddings with darker colors as subgoals could lead the agent to expand the explored areas.

Algorithm 1 provides the full procedure of the proposed approach, HESS. To balance exploration and exploitation, HESS probabilistically utilizes the exploration strategy in Equation 6 or explores with the learned policy $\pi_h$ (Line:5-6), and the probability $p$ is decayed over the course of learning. The representation $\phi$ is updated every $I$ episodes (Line:13-15) so that $\phi$ is stable during the update interval, and the proposed regularization is to maintain the stability before and after the update.

---

**Algorithm 1** HESS algorithm

---

 1: **Initialize:** $\pi_h(g|s)$, $\pi_l(a|s^l, g)$ and $\phi(s)$.
 2: **for** $i = 1..num\_episodes$ **do**
 3:     **for** $t = 0..T - 1$ **do**
 4:         **if** $t \equiv 0 \pmod{c}$ **then**
 5:             With a probability of $p$, explore with the strategy in Eq. 6.
 6:             With a probability of $1 - p$, sample subgoal $g_t$ with learned policy $\pi_h$.
 7:             Update $\pi_h$ with an off-policy RL method.
 8:         **else**
 9:             $g_t = g_{t-1}$
10:         **end if**
11:         $r_t, s_{t+1} \leftarrow$ execute $a_t \sim \pi_l(\cdot|s_t^l, g_t)$, and update $\pi_l$ with an off-policy RL method.
12:     **end for**
13:     **if** $i \equiv 0 \pmod{I}$ **then**
14:         Update $\phi$ with Eq. 3 using $m$ minibatches.
15:     **end if**
16: **end for**
17: **Return:** $\pi_h, \pi_l$ and $\phi$.

---

## 4 RELATED WORK

Learning effective subgoal representations has been an important and challenging problem in GCHRL (Dwiel et al., 2019). Previous works have proposed to learn the subgoal representation in an end-to-end manner with policies (Vezhnevets et al., 2017; Dilokthanakul et al., 2019), or by bounding the sub-optimality of the hierarchical policy (Nachum et al., 2019a), or with a learning objective of slow dynamics (Li et al., 2021). Nevertheless, those methods have not considered the stability of the subgoal representation learning, which results in a non-stationary high-level learning

environment. Other prior methods utilize a predefined or pretrained subgoal space (Péré et al., 2018; Nair & Finn, 2019; Zhang et al., 2020; Ghosh et al., 2018) to keep the stability of the subgoal representation. However, those methods require task-specific human knowledge or extra training data. In this work, we propose a novel regularization to stabilize the subgoal representation learning. Specifically, the subgoal representation is learned with a contrastive triplet loss. The contrastive objective is used as an auxiliary representation loss in other RL literature as well (Oord et al., 2018; Laskin et al., 2020; Fortunato et al., 2019).

Benefiting from temporally extended exploration, HRL methods have shown better exploration abilities (Nachum et al., 2019b). Bottom-up HRL works learn a set of diverse skills or options in a self-supervised manner, and use those semantically meaningful low-level skills to explore in downstream tasks (Jinnai et al., 2020; Co-Reyes et al., 2018; Eysenbach et al., 2018; Sharma et al., 2019; Li et al., 2019). Nevertheless, the skills produced by those methods may not be required by the downstream tasks, since when learning the skills, the agent knows nothing about the downstream task rewards. In GCHRL, Zhang et al. (2020) restricted the high-level action space to a $k$-step adjacent region of the current state to achieve better sample efficiency. Röder et al. (2020) aimed at improving high-level exploration via curiosity-driven intrinsic rewards. However, those methods require oracle subgoal spaces designed with prior knowledge. In contrast, HESS learns subgoal representations online. Li et al. (2021) proposed an HRL approach, LESSON, with online subgoal representation learning, and provided theoretical analysis that the learned representation could support efficient exploration. However, without intrinsic rewards at the high level, LESSON could hardly solve long-horizon tasks with extremely sparse rewards. This paper improves exploration from an orthogonal perspective from LESSON, and develops an active exploration strategy for high-level policy learning.

We propose two measures for subgoals: novelty and potential. The cumulative count in the novelty measure is related to successor features (Kulkarni et al., 2016b). Previous exploration methods use successor features to propagate the uncertainty of value function (Janz et al., 2019) or whether the features occur (Machado et al., 2020). In contrast, our method propagates an explicit estimation of the number of feature occurrence through trajectories, which is conducted without function approximation. The potential measure of reachability shares some similarities with empowerment (Salge et al., 2014), which is defined as the agent's control over its environment. By maximizing empowerment (Campos et al., 2020; Gregor et al., 2016; Mohamed & Rezende, 2015), an agent could reach more diverse states. Unlike empowerment measured by mutual information, the potential measure is formalized with the distances between desired subgoals and achieved latent states. In Appendix C, we discuss how HESS is related to goal selection strategies in the multi-goal RL domain (Schaul et al., 2015).

## 5 EXPERIMENTS

We evaluate HESS on a set of challenging sparse-reward environments that require a combination of locomotion and object manipulation skills, aiming at answering the following questions: (1) Can HESS outperform state-of-the-art exploration strategies in sample efficiency and overall performance? (2) Can HESS learn a stable subgoal representation space efficiently? (3) How important are the various components of the HESS agent? (4) What do the subgoals selected by HESS look like? (5) How much would the choice of hyper-parameters influence the experimental performance? For better sample efficiency, we utilize an off-policy algorithm, Soft Actor-Critic (SAC) (Haarnoja et al., 2018), to learn both-level policies. To mitigate the influence of the entropy term of SAC on HESS, we use SAC with automatic entropy tuning.

### 5.1 ENVIRONMENT SETUP

We evaluate on a suite of MuJoCo (Todorov et al., 2012) tasks that are widely used in the HRL community, including Ant (or Point) Maze, Ant Push, Ant FourRooms, Cheetah Hurdle, Cheetah Ascending, and two variants with low-resolution image observations. The experiments with image input are labeled 'Images'. To demonstrate the exploration ability of HESS, we adapt those tasks to make them more challenging. Different from the settings of random start or random goal with dense external rewards (Nachum et al., 2018; 2019a; Li et al., 2021; Zhang et al., 2020), in the tasks used in this work, a simulated robot starts from a fixed position and needs to reach a faraway target position

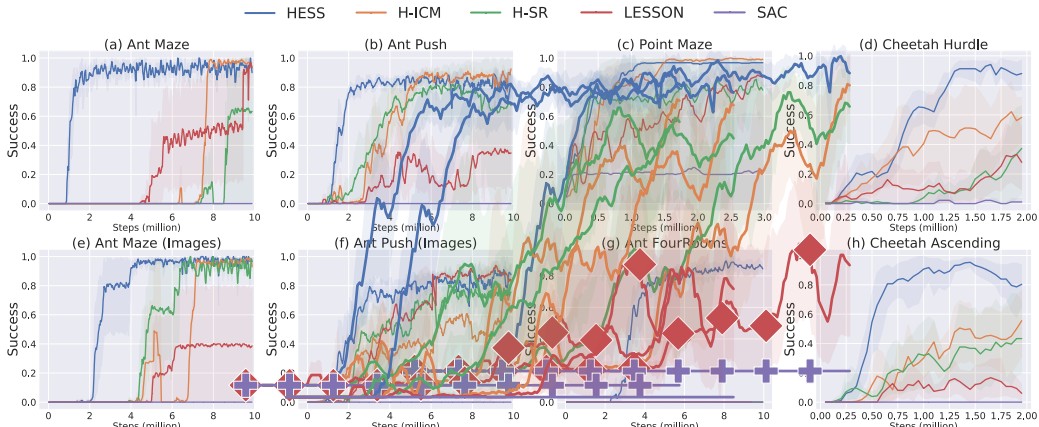

Figure 4: Learning curves of the proposed method and baselines on all the tasks. The $y$ axis shows the average success rate in 10 episodes. Each line is the mean of 5 runs with shaded regions corresponding to a confidence interval of 95%. All the curves have been smoothed equally for visual clarity. Code is available at https://github.com/SiyuanLee/HESS.

with sparse external rewards. Furthermore, there is no predefined subgoal space provided. More details about the environments and implementation are available in Appendix A and B, respectively.

## 5.2 COMPARATIVE ANALYSIS

We conduct experiments comparing the proposed hierarchical exploration approach to the state-of-the-art baselines: (1) *H-ICM* (Pathak et al., 2017): prediction error in the latent subgoal space as intrinsic rewards to the high level. (2) *H-SR* (Machado et al., 2020): count-based exploration bonus in the high level, and the counts are estimated by the norm of the successor representation. (3) *DADS* (Sharma et al., 2019): an unsupervised skill discovery method via predicting dynamics. (4) *LESSON* (Li et al., 2021): an GCHRL method learning the subgoal representation online using triplet loss, and no intrinsic rewards in the high level. (5) *SAC* (Haarnoja et al., 2018): the non-hierarchical base RL algorithm used in this work.

As shown in Figure 4, the proposed method substantially outperforms the baselines in all the tasks. The outperformance in the Ant FourRooms task is more significant since the maze scale of this task is larger, and the exploration problem is harder. Both H-ICM and H-SR underperform our method. The dynamic model of the simulated physics engine is relatively easy to fit by neural networks, so the intrinsic rewards of H-ICM may vanish before the policies have been learned well. The H-SR method estimates visit counts using the $\ell_1$-norm of the successor representation, which is the expected discounted future state occupancies in trajectories. This idea shares some similarities with the cumulative counts in the novelty measure. Nevertheless, the successor representation estimates the expected future state occupancy starting from a given state (Dayan, 1993), but not the visitation number of the given state, which is less helpful to promote exploration.

When predicting dynamics in the whole observation space, the intrinsic rewards of DADS could hardly help the agent learn gait skills that make the robots move. The original paper (Sharma et al., 2019) has also demonstrated that without $x, y$ prior, the trajectories generated by a primitive skill have a large variance. We provide a visualization of those skills in Appendix D. As a result, the success rates of DADS on all the tasks are zero. Hence, we omit them from Figure 4. LESSON uses no intrinsic reward to promote exploration in the high level. Even with a good subgoal representation space, it still could not solve those tasks with sparse rewards. The non-hierarchical method SAC performs poorly in all the tasks, demonstrating the strength of the hierarchical structures in solving long-horizon tasks with sparse rewards.

## 5.3 VISUALIZATION OF REPRESENTATION LEARNING

We visualize state embeddings of 5 trajectories from the beginning to the end of the $\supset$-shape corridor in the Ant Maze (Images) task in Figure 6, comparing the representations learned by HESS and HESS without stability regularization. Those representations are all learned from top-down image observations.

Figure 5: Ant Maze

HESS is able to learn an effective subgoal representation at an early stage. By optimizing the contrastive objective, the Euclidean distances in the latent space approximately cor-

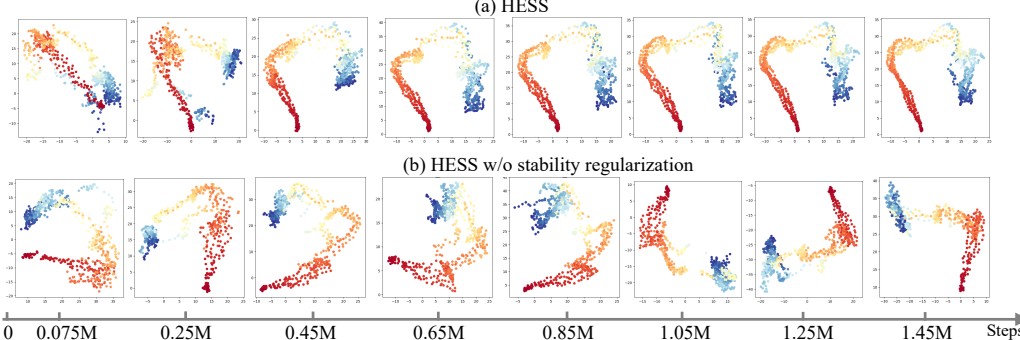

Figure 6: Subgoal representation learning process in the Ant Maze (Images) task. Each subfigure contains 2D state embeddings of 5 trajectories in the ⊃-shape maze (red for the start, blue for the end). For larger axis labels, see videos of the representation learning process at `https://sites.google.com/view/hess-iclr`.

respond to the number of transitions between states in local areas, which provides subgoal-reaching rewards for low-level policies. But globally, the distance could not represent the number of transitions. For example, it requires lots of transitions to move from the start to the goal in the ⊃-shaped maze, but they have not been pushed much away in the latent space, since there is no explicit constraint in the triplet loss for the distance between features more than $c$ timesteps apart. Comparing Figure 6(a) to Figure 6(b), we find that with the stability regularization, when the features have already fitted the learning objective well, their changes are minor afterward. There is almost no change for the red features from 0.25M steps to the end in Figure 6(a).

Without the stability regularization, all the features are changing dramatically. The non-stationary issue makes high-level policy learning very hard. In Figure 6(b), at 0.25M steps (the second subfigure), the high-level agent at the red start should select a subgoal to move upwards. However, at 0.65M steps (the fourth subfigure), selecting a subgoal in the right of the start state is better.

## 5.4 ABLATIVE ANALYSIS OF VARIOUS COMPONENTS

To understand the importance of various design choices of HESS, we conduct ablation studies on the subgoal representation learning and the proposed exploration strategy separately. In the ablation studies of representation learning, we compare HESS, HESS without stability regularization, and HESS without both stability regularization and prioritized sampling, shown by the curves without markers in Figure 7. In the tasks with image input, the advantage of stability regularization and prioritized sampling is more substantial, since the representation of image observations is harder to learn. In the tasks with vector input, the subgoal representation has a good generalization ability, so the effect of stability regularization and prioritized sampling becomes less significant.

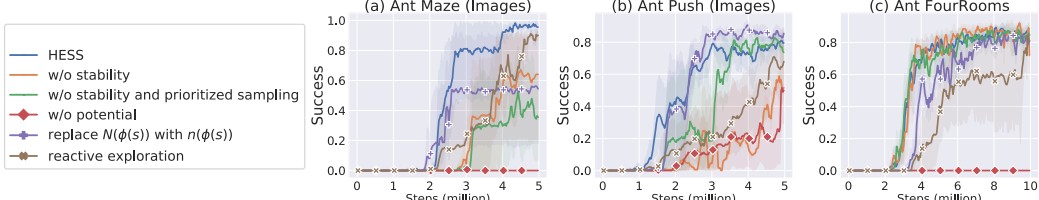

Figure 7: Ablation studies of representation learning and the exploration strategy.

In the ablation studies of the exploration strategy, we compare HESS, HESS without the potential measure, replacing cumulative counts $N(\phi(s))$ with immediate counts $n(\phi(s))$, and reactive exploration with high-level intrinsic rewards $r_i(s) = \eta_1/\sqrt{N(\phi(s))} + \eta_2/\sqrt{-U(\phi(s))}$, shown by the curves with markers in Figure 7. The potential measure is extremely important, since without it, the subgoals are concentrated on the areas with low counts, but not extendable, like the areas near the corners in Figure 8(b). Therefore, the success rates without the potential measure are much lower.

With immediate counts $n(\phi(s))$ as the novelty measure, the agent loses a long-term vision to seek out novel subgoals, and only focuses on novel states in its neighborhood. Hence, the states near the walls are selected as subgoals more frequently in Figure 8(c) than in Figure 8(a), but those subgoals are still better for exploration than the subgoals selected without the potential measure. In the ablation study of active exploration, we can see that by actively selecting favorable subgoals, HESS has achieved better sample efficiency than the reactive exploration method with intrinsic

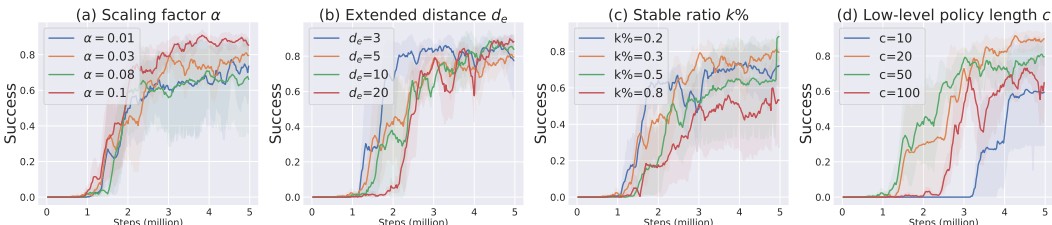

Figure 8: Visualization of the selected subgoals during 10 episodes and their corresponding positions in the $x, y$ space in the Ant Maze task at $0.25$ million timesteps.

rewards, as the reactive method could not immediately seek the novel and promising subgoals even when it has detected them.

### 5.5 ABLATION STUDIES ON HYPER-PARAMETER SELECTION

In this section, we set up a set of ablation tests on several hyper-parameters of HESS in the Ant Push (Images) task. From Figure 9, we can see that most parameters work in a large range. Furthermore, we use a single suite of parameters in all the tasks in this paper, which also indicates that the proposed approach is robust to hyper-parameter selection.

Figure 9: Ablation studies of hyper-parameter selection in the Ant Push (Images) task.

**Scaling factor** $\alpha$ balances novelty and potential. Smaller $\alpha$ encourages HESS to choose more novel state embeddings as subgoals. Results in Figure 9 (a) indicate that HESS is robust against $\alpha$, since the proposed method could work well, when the potential measure mitigates the negative influence of novelty inaccuracy on exploration. For all tasks in Section 5.2, we set $\alpha = 0.03$.

**Extended distance** $d_e$ is the distance between the subgoal selected from the replay buffer and the imagined subgoal $g_e$. To keep $g_e$ still near the current latent state, $d_e$ should be no larger than the radius $r_g$ of the neighborhood for selecting subgoals. The performance of HESS is reasonable as long as $d_e$ is not too large, as an excessively large $d_e$ may cause the subgoal $g_e$ pursued by the low level too far away from the current latent state. For all tasks in Section 5.2, we set $d_e = 5.0$.

**Stable ratio** $k\%$**:** The stability regularization constrains the changes of the embeddings for $k\%$ of the states with the minimum triplet loss in the buffer. When $k\%$ is too large, the representation learning efficiency may be slightly affected, which is harmful to the hierarchical policy learning, as indicated in Figure 9 (c). For all tasks in Section 5.2, we set $k\% = 0.3$, i.e., the regularization is applied to $30\%$ of the data in buffer.

**Low-level policy length** $c$ is an important and common hyper-parameter in HRL. With a larger $c$, the burden of high-level decision-making is lighter, but low-level policy learning becomes harder. The learning performance is better when $c$ is 20 or 50 with an episode length of 500. Among all the hyper-parameters, $c$ seems to influence the performance most. For all tasks and baselines in Section 5.2, we set $c = 50$.

## 6 CONCLUSION

To solve long-horizon sparse-reward tasks with GCHRL, we design novelty and potential measures for subgoals upon stable subgoal representation learning, and develop a hierarchical exploration strategy that actively seeks out new promising subgoals and states. As the dimension of the subgoal space in this work is low, we employ a naive count estimation method in the representation space. When the dimension of the subgoal space is higher, it would be better to learn a density model (Papamakarios et al., 2019), or utilize more advanced count estimation methods, such as kernel-based methods (Davis et al., 2011). For the future work, we believe that the idea of stability is general and could be used beyond HRL. For example, maybe many existing techniques in non-deep RL could be applied to representations learned with stability as well.

## ACKNOWLEDGEMENT

This work is supported by Science and Technology Innovation 2030 – "New Generation Artificial Intelligence" Major Project (No. 2018AAA0100904) and National Natural Science Foundation of China (No. 20211300509).

## REPRODUCIBILITY STATEMENT

For reproducibility, we include an anonymous downloadable source code in the supplementary material. Beyond that, we describe the details about the environments and the implementation in Appendix A and B.

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

## A    ENVIRONMENT DETAILS

The environments of Point Maze, Ant Maze, Ant Push, Ant FourRooms, Cheetah Hurdle, and Cheetah Ascending are shown in Figure 10. The environment rewards are sparse and binary. We define a "success" as being within an L2 distance of 1.5 from the goal. If the agent successfully reaches the goal, the reward is 1, and else the reward is 0. The observation space includes the current position and velocity of the simulated robot and the environment goal. For the 'Images' versions of these environments, we zero-out the $x, y$ coordinates in the observation and append a low-resolution $5 \times 5 \times 3$ top-down view of the environment, as described in (Nachum et al., 2019a; Li et al., 2021). Each episode ends at 500 timesteps for the Ant Maze, Point Maze, and Ant Push task and ends at 1000 timesteps for the Ant FourRooms, Cheetah Hurdle, and Cheetah Ascending task.

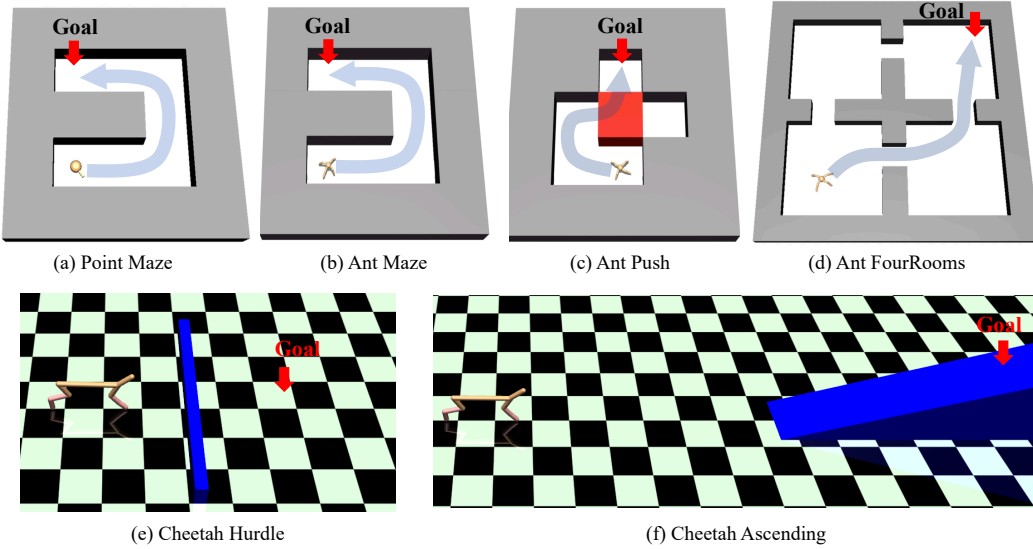

| (a) Point Maze | (b) Ant Maze | (c) Ant Push | (d) Ant FourRooms |

| (e) Cheetah Hurdle | (f) Cheetah Ascending |

Figure 10: A collection of benchmark sparse-reward environments.

**Ant Maze.** The size of the ⊃-shape maze is $12 \times 12$. For both training and evaluation, the start position is fixed to (0, 0), and the environment goal is fixed to (0, 8).

**Point Maze.** The maze structure is the same as the Ant Maze task. The simulated robot is a point mass, which is controlled by actions of dimension two, rotation on the pivot of the point mass and movement in the direction of the pivot.

**Ant Push.** In this task, a movable block is placed at (0, 4). The agent is initialized at position (0, 0), and the goal position is fixed to (0, 8). Therefore, the agent must first move to the left, then push the movable block to the right, and then navigate to the goal unimpeded.

**Ant FourRooms.** In this environment, the maze structure is larger, $18 \times 18$. The start position is (0, 0), and the goal is (14, 14).

**Cheetah Hurdle.** The simulated Half-Cheetah robot is required to stride over a hurdle with a height of 0.35 and walk to the goal, which is 3.2 meters from the start position.

**Cheetah Ascending.** The simulated Half-Cheetah robot has to climb up to a hill to reach the goal. The height of the goal position is 1.

## B    IMPLEMENTATION DETAILS

Except for the hyper-parameters discussed in Section 5.5, we list other hyper-parameters and their ranges considered in the experiments in Table 1. The probability $p$ of actively selecting subgoals with the proposed exploration strategy is initialized as 0.7, and linearly annealed to 0 at half of the total training steps. For subgoal selection, our method firstly samples $M$ states satisfying the constraint

from the replay buffer $\mathcal{B}$, $\{s_i | D(\phi(s_i), \phi(s_t)) < r_g\}_{i=1}^{M} \sim \mathcal{B}$, and then selects a favorable subgoal with Equation 6 from the candidate subgoal set $\{\phi(s_i)\}_{i=1}^{M}$. When calculating the expectation in the novelty and potential measures (Eq. 4 and 5), to deal with the off-policyness of the data in the buffer, we decrease the effects of the older rollouts with a discount factor $\gamma = 0.995$, taking inspiration of the Exponential Moving Average (EMA) method.

For the neural network structures, the actor network for each level is a Multi-Layer Perceptron (MLP) with two hidden layers of dimension 256 using ReLU activations. The critic network structure for each level is identical to that of the actor network. We scale the outputs of the actor networks of both levels to the range of corresponding action space with tanh nonlinearities. The representation function $\phi(s)$ is parameterized by an MLP with one hidden layer of dimension 100 using ReLU activations. We use PyTorch and Adam optimizer to train all the networks. Experiments are carried out on NVIDIA GTX 2080 Ti GPU. The learned hierarchical policies are evaluated every 25000 timesteps by averaging performance over 10 episodes.

Table 1: Hyper-parameters for experiments

| Hyper-parameters | Values | Ranges |
|---|---|---|
| Subgoal dimension $d$ | 2 | |
| Radius $r_g$ of subgoal selecting neighborhood | 20 | $\{10, 20\}$ |
| Grid size for count estimation | 3 | $\{2, 3, 5, 8\}$ |
| Number of candidate subgoals $M$ | 1000 | |
| Learning rate for both level policies | 0.0002 | |
| Learning rate for the subgoal representation | 0.0001 | |
| Discount factor $\gamma$ for both levels and cumulative counts | 0.99 | |
| Soft update rate for both levels | 0.005 | |
| Replay buffer size for both levels | $1e6$ | |
| Reward scaling for high level | 0.1 | $\{1, 0.1\}$ |
| Reward scaling for low level | 1 | $\{1, 0.1\}$ |
| High-level action frequency $c$ | 50 | $\{10, 20, 50, 100\}$ |
| Batch size for both level policies | 128 | |
| Batch size for representation learning | 100 | |
| Intrinsic reward coefficient $\eta_1$ | 1000 | $\{10, 100, 1000\}$ |
| Intrinsic reward coefficient $\eta_2$ | 10 | $\{10, 100, 1000\}$ |
| Stability regularization coefficient $\lambda_0$ | 0.1 | |
| Number of episodes $I$ between representation updates | 100 | |
| Number of minibatches $m$ in one representation update | 50000 | $\{10000, 50000\}$ |

## C RELATED WORK

With the proposed novelty and potential measures, we develop a subgoal selection strategy to improve exploration in HRL. There are numerous goal selection strategies in the multi-goal RL domain (Schaul et al., 2015) as well, including sampling diverse goals uniformly from a buffer (Warde-Farley et al., 2018; Pong et al., 2019), sampling goals from the achieved goal distribution (Nair et al., 2018), sampling goals of intermediate difficulty from a generative model (Florensa et al., 2018), and selecting goals in sparsely explored areas (Pitis et al., 2020). However, those methods use predefined or pretrained goal spaces. In addition, unlike HESS, those flat goal-conditioned methods pursue only one goal in an episode and thus may be less able to decompose complex tasks.

## D BASELINES

**H-SR** is the baseline method using count-based intrinsic rewards in the high level, where the counts are estimated with the $\ell_1$-norm of the successor representation (Machado et al., 2020). Since the $\ell_1$-norm of the successor representation only counts for whether the features occur in trajectories, but not the times of feature occurrence. Therefore, the intrinsic rewards of H-SR vanish at an early learning stage. We show how the variance of the intrinsic rewards changes with the environment

timesteps in the Ant Maze task in Figure 11. Since there is little difference in the intrinsic rewards among different states, the H-SR intrinsic rewards could hardly facilitate efficient exploration in a late learning stage.

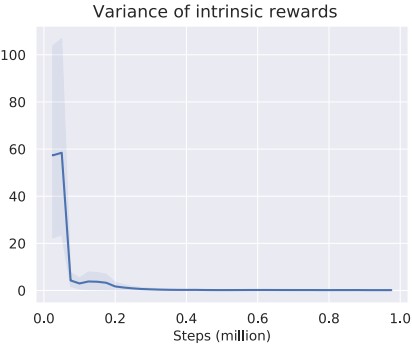

Figure 11: The variance of the intrinsic rewards of the H-SR method in the Ant Maze task, with x-axis as millions of environment timesteps.

**DADS** learns a set of primitive skills by predicting the skill dynamics and then uses the learned skills in downstream tasks (Sharma et al., 2019). The intrinsic rewards of skill learning encourage a skill to produce transitions (a) predictable under the dynamic model and (b) different from the transitions produced under other skills. However, when the dynamic model is defined on the whole observation space without the $x, y$ prior, the trajectories of the learned skills suffer from a large variance, as shown in Figure 12. Those high variance skills offer limited utility for hierarchical control.

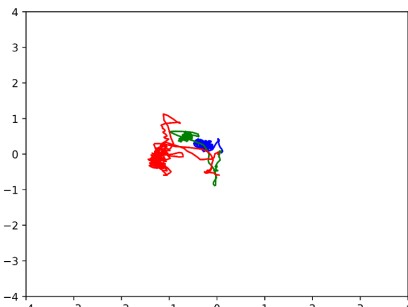

Figure 12: X-Y traces of skills learned by DADS in the Ant Maze task, where the same color represents trajectories resulting from the execution of the same skill.

**FeUdal Networks (FuN)** (Vezhnevets et al., 2017) also utilizes directional subgoal vectors. We have run this method in the Ant Maze task for 70 million steps, and the success rate of FuN is still zero. We analyze the reason for the underperformance of FuN, and find that the subgoal representation learned by FuN is much worse than ours, as shown in Figure 13, since the representation learning in FuN lacks a strong inductive bias. In our approach, we use the contrastive objective to learn the subgoal representation.

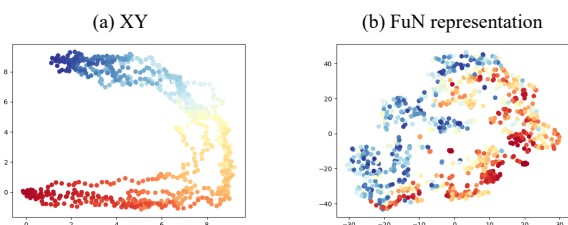

Figure 13: Trajectories in the XY space and the FuN representation space. The dimension of the FuN representation space is 50, and we use t-SNE to perform dimension reduction and visualize the latent trajectories.

# E  ADDITIONAL EXPERIMENTAL RESULTS

## E.1  ABLATION STUDIES ON ADDITIONAL HYPER-PARAMETERS

We conduct ablation studies on additional hyper-parameters in the Ant Maze and Ant Maze (Images) task, and find that the representation update interval $I$ which influences the representation stability as well, is also important to learning performance.

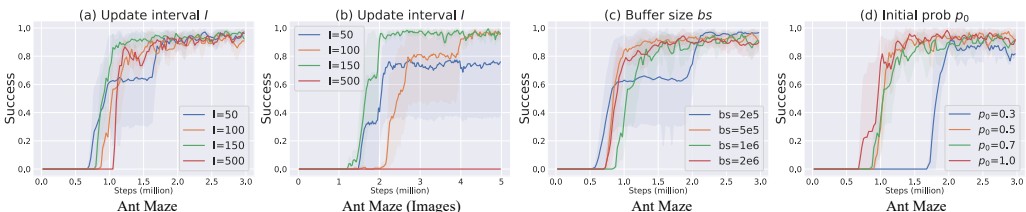

Figure 14: Ablation studies on additional hyper-parameters.

**Representation update interval** $I$ (episodes) balances the stability and efficiency of the representation learning. The task performance is robust when $I$ is in a proper range ($50 \leq I \leq 150$). With a short update interval ($I = 50$), the learning performance is slightly hurt since the subgoal representation is less stable. When $I$ is too large ($I = 500$), a successful policy could hardly be learned in the Ant Maze (Images) task, as the large update interval hinders the representation learning efficiency.

**Buffer size** $bs$ for both levels. Figure 14 demonstrates that HESS is indeed robust against this hyper-parameter when $2e5 \leq bs \leq 2e6$.

**Initialization** $p_0$ for probability $p$. When $p_0$ is set to a relatively large value ($p_0 \geq 0.5$), HESS achieves stable performance, as shown in Figure 14. Small $p_0$ (0.3) makes the proposed exploration strategy seldom used, which leads to poor sample efficiency.

## E.2  ABLATION STUDY ON IMAGINED SUBGOALS

To guide the low-level controller to reach unexplored areas, we firstly sample state embedding $g_t$ from the neighborhood of the current latent state with a radius of $r_g$, and then extend $g_t$ with an extended distance $d_e$ to obtain an imagined subgoal $g_e$. In this section, we compare the performance of directly increasing $r_g$ and perturbing $g_t$ in subgoal selection.

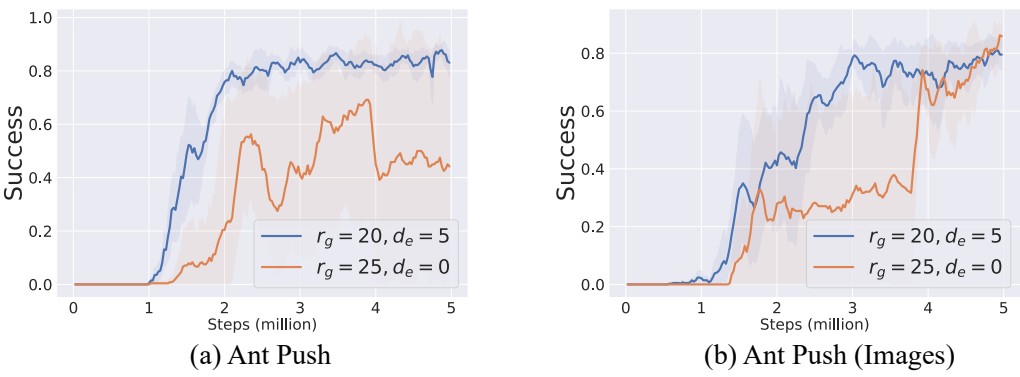

Figure 15: Ablation studies on the imagined subgoals in Ant Push and Ant Push (Images) task.

Results in Figure 15 demonstrate that when $d_e > 0$, the sample efficiency is better. $g_e$ is obtained via perturbing an achieved subgoal, which might be in the unexplored areas, and thus more helpful to exploration. Only increasing $r_g$, the sampled subgoal $g_t$ is still be in explored areas, which is less useful to expand the explored areas.

### E.3 NOVELTY ESTIMATION

We estimate the novelty measure $N(\phi(s))$ with the data in the replay buffer. To make the novelty estimation more accurate, we need to deal with the off-policyness of the data in the buffer, so we adopt the EMA method to decrease the effect of old rollouts, as described in Appendix B. An alternative way to deal with the off-policyness is the importance sampling (IS) technique, i.e., $N(\phi(s_i)) = \mathbb{E}_{\mathcal{B}}[\sum_{j=0}^{\lfloor (T-i)/c \rfloor} \gamma^j \rho^{is} n(\phi(s_{i+jc}))]$, where $\rho^{is} = \prod_{t=0}^{(i+jc)/c} \pi_{new}^h(g|s_{tc})/\pi_{old}^h(g|s_{tc})$, $\pi_{new}^h$ is the current high-level policy, $\pi_{old}^h$ is the old high-level sampling policy at state $s_{tc}$, $\mathcal{B}$ is the replay buffer. We compare these two implementations in the Ant Maze task, and those two methods empirically demonstrate comparable performance, as shown in Figure 16.

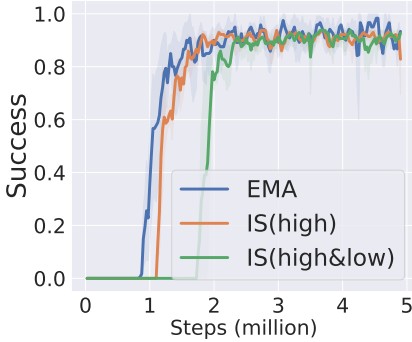

Figure 16: Different novelty estimation in the Ant Maze task.

### E.4 IMPLEMENTATION OF THE STABILITY REGULARIZATION

In this section, we ablate different implementations of the weighting function $\lambda(s)$ in the stability regularization:

- Binary $\lambda(s)$: For the top $k\%$ of the triplets with the minimum representation losses, we set $\lambda(s) = \lambda_0$ for the anchor states ($s_t$) in these triplets, and for the other states, $\lambda(s) = 0$.
- Continuous $\lambda(s)$: Weighting $\lambda(s)$ continuously based on the representation loss, i.e., $\lambda(s_t) = \lambda_0/\sqrt{1 + L_{tri}(s_t, \cdot, \cdot)}$, where the triplet $(s_t, \cdot, \cdot)$ is uniformly sampled from the replay buffer.

Compared to the *continuous* $\lambda(s)$ implementation, *binary* $\lambda(s)$ sharply distinguishes the well-optimized embeddings from the underfitting ones, and thus has a slightly better performance, as demonstrated in Figure 17.

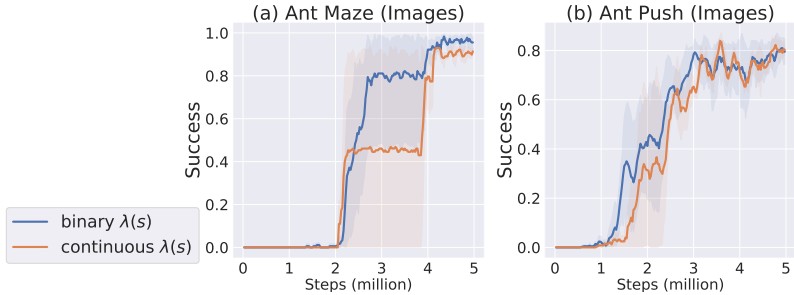

Figure 17: Different implementations of weighting function $\lambda(s)$.

