# OpenReview forum: "Active Hierarchical Exploration with Stable Subgoal Representation Learning"
_ICLR.cc/2022/Conference — ICLR 2022 Poster_

### Official Review · Reviewer_EgA9 · 2021-10-28

**Correctness:** 4
**Technical Novelty And Significance:** 3
**Empirical Novelty And Significance:** 4
**Recommendation:** 8
**Confidence:** 4

**Main Review:**

This paper is really well executed. It builds on top of an already complicated architecture adding more than one new component to that architecture, but it does so while providing proper intuitions for each one of these new components and, more importantly, actually doing ablation studies that quantify the impact of each component. To me, Section 5.4 is the highlight of the paper. I also appreciated Section 5.5, which shows how the paper is also concerned with stability over different parameters introduced by the proposed metric. I think the paper would benefit from further clarifying some parts of the text, but otherwise this is a good paper. Specifically:

* In the Introduction, for example, it is said that methods based on visit count are "_reactive_ exploration methods that need to learn how to maximize the intrinsic rewards before performing exploratory behaviors". I don't necessarily disagree with that, although the whole idea of visit counts is to incentivize these exploratory behaviors. My question though, is: isn't this exactly the same with the proposed idea? It does use counts and not only that, but also expected state visitation counts for the trajectory, which is even more demanding in terms of having to visit the state first.

* In Section 2, when defining $U(\phi(s_t))$, the distance is defined to be between $g_t$ and $\phi(s_t)$. For the proposed algorithm, should it be $g_e$ instead of $g_t$? Still on Section 2, it is said "we concatenate states and state embeddings together as the low-level states". What does this actually mean? What are the states here? For images, for example, would it literally be all pixels on the screen?

* In Section 3.1, I don't think $\lambda_0$.

* In Section 3.2, it is said "low-dimensional continuous latent space into discrete cells, i.e., the state embeddings are mapped into cells containing them.". What are these cells? How were they defined? I can imagine this is somewhat straightforward to do if you assume you have access to x,y positions, but how is this done in higher dimensional settings? How are these cells defined for Images, for example?

* In Section 3.2, when discussing the potential measure, it is said that Figure 3 demonstrates that "with online representation learning, the novelty measure is a mixture of counts in the past and current representation spaces, so it might mislead the exploration". How is that? I couldn't understand what I should be looking at in Figure 3 to reach this conclusion.

* In Section 3.3, it is said that the active exploration strategy "avoids the non-stationary issue". How? Aren't these reward signals changing constantly based on counts and the representation being learned? How does the active exploration strategy actually avoids the non-stationarity issue?

* In Section 4, it is said that "Bottom-up HRL works learn a set of diverse skills or options in a self-supervised manner, and use those semantically meaningful low-level skills to explore in downstream tasks", but "those methods may produce some redundant and useless skills". This claim is not backed up by any reference or experiment. Why is this true when some of these methods explicitly ask for diverse skills that are not supposed to overlap to each other?

* In Figure 4, how were the confidence intervals computed if only 5 samples were available?

* In Section 5.2 it is said "the successor representation estimates the expected future state occupancy starting from a given state (Kulkarni et al., 2016b), but not the visitation number of the given state, which is less helpful to promote exploration." However, isn't this exactly what H-SR shows? That the $\ell_1$ norm of the SR captures the visitation number of a given state? Moreover, the reference to the SR should be "Peter Dayan: Improving Generalization for Temporal Difference Learning: The Successor Representation. Neural Comput. 5(4): 613-624 (1993)".

* No details were given on how Figure 6 was generated. I don't know how to reproduce it.

* *Importantly, in the ablations, were the parameters of the ablated methods tuned?*


**Summary Of The Paper:**

This paper proposes a new algorithm for goal-conditioned hierarchical reinforcement learning that is able to succeed in tasks with sparse rewards (differently from most other methods in the field). It does so through two innovations: (1) a representation learning procedure that is more stable, and (2) a exploration strategy that takes into consideration not only novelty but also reachability.

Specifically, the representation learning procedure is based on what is now a standard a contrastive loss, but it is augmented by a regularization term that make the learning procedure stable where the representation is already satisfactory, allowing goal sampling to be more effective. Figure 6 is a particularly nice visualization of the impact of this regularization term.

The exploration strategy to sample goals to be visited is also novel. Instead of using goal visitation counts, this paper proposes the idea of using expected sum of state visitation counts from that state onwards, capturing some notion of long term novelty. Moreover, the exploration bonus also has a potential term that captures how promising each goal state is in terms of how far from the goal state the agent is expected to end up. Quantitative impact is reported in Figure 7, but I particularly liked the intuitions/visualizations provided in Figure 8.

**Summary Of The Review:**

This paper is really well executed. It builds on top of an already complicated architecture adding more than one new component to that architecture, but it does so while providing proper intuitions for each one of these new components and, more importantly, actually doing ablation studies that quantify the impact of each component. To me, Section 5.4 is the highlight of the paper. I also appreciated Section 5.5, which shows how the paper is also concerned with stability over different parameters introduced by the proposed metric. I think the paper would benefit from further clarifying some parts of the text, but otherwise this is a good paper.

---

> ### Author Response · Authors · 2021-11-17
> **Response to Reviewer EgA9 (Part 1)**
>
> Thank you for the encouraging and detailed comments!
>
> **Q1.** "*reactive* exploration methods that need to learn how to maximize the intrinsic rewards before performing exploratory behaviors". Isn't this exactly the same with the proposed idea?
>
> **A1.** Given high-quality intrinsic rewards, reactive methods could not **immediately** perform exploratory behaviors, since learning how to maximize the intrinsic rewards requires large amounts of samples as well. When the parametric policies of the reactive methods are well optimized, then they have good exploration abilities. In contrast, with a proper estimation of the visit counts, our approach could immediately perform exploratory behaviors, since the proposed measures directly take effects on the non-parametric active exploration policy. Empirical results in Section 5.4 also show that our active exploration leads to more efficient learning than reactive exploration methods.
>
> **Q2.** **(1)** $U(\phi(s_t))$ in Section 2. **(2)** "we concatenate states and state embeddings together as the low-level states". What does this actually mean?
>
> **A2.** **(1)** Thank you for pointing out this typo. The $U(\phi(s_t))$ in Section 2 denotes the neighborhood of the current latent state, which has nothing to do with the potential measure defined in Section 3. We have revised the notation representing the neighborhood in Section 2.
>
> **(2)** The low-level policy is conditioned on both the state provided by the environment and the corresponding learned state embedding. Practically, in image-version tasks, the latent embedding $\phi(s)$ is concatenated to the hidden layer in the low-level policy network.
>
> **Q3.** In Section 3.1, I don't think $\lambda_0$.
>
> **A3.** $\lambda_0$ is a positive parameter, which helps stabilize the well-optimized state embeddings. If the reviewer has further questions about $\lambda_0$, please let us know.
>
> **Q4.** What are these cells in Section 3.2? How were they defined?
>
> **A4.** We use a mesh to discretize the learned low-dimensional representation space, and then those discrete embeddings are projected into a finite hash table.
>  Note that the visitation number is not counted in the x,y position space, or the state space, but it is defined in the learned subgoal space, and thus our approach could solve high-dimensional tasks with image inputs.
>
> **Q5.** "with online representation learning, the novelty measure is a mixture of counts in the past and current representation spaces, so it might mislead the exploration". What to look at in Figure 3 to reach this conclusion?
>
> **A5.** The black arrows in Figure 3(c) show the representation changes. Due to online representation learning, the updated representation function produces new embeddings different from previous ones for the same batch of states. Therefore, counting the mixed embeddings is inaccurate. For example, although the latent density in the red box is low, the corresponding area in the x,y space is not actually less explored.
>
> **Q6.** How does the active exploration strategy avoid the non-stationary issue?
>
> **A6.** In our active exploration method, the changing novelty and potential do not serve as intrinsic rewards, and thus would not affect the policy optimization. Instead, with the proposed measures, the active exploration strategy directly guides the behavior policy to efficiently explore novel and reachable states.
>
> **Q7.** In Section 4, the claim of bottom-up HRL works is not backed up.
>
> **A7.** Our previous narrative about bottom-up HRL methods was not precise. We intended to express that in bottom-up HRL, skills are learned in an unsupervised manner, so these methods may learn skills unrelated to downstream tasks.
>
> In our approach, the learning of each low-level skill is necessary, since the low-level intrinsic rewards are induced by the subgoals selected by the high-level policy, and the high-level policy is trained with task rewards. In summary, the relationship between high-level and low-level learning in our method is tighter.
>
> We have modified the claim about bottom-up HRL methods in Section 4.
>
> **Q8.** In Figure 4, how were the confidence intervals computed?
>
> **A8.** We used the [`lineplot`](https://seaborn.pydata.org/generated/seaborn.lineplot.html) function with the default arguments in the `seaborn` package to plot confidence intervals in Figure 4.

---

> > ### Author Response · Authors · 2021-11-17
> > **Response to Reviewer EgA9 (Part 2)**
> >
> > **Q9.** Does the $\ell_1$ norm of the SR capture the visitation number of a given state?
> >
> > **A9.** No, Theorem 1 in [1] proves that the $\ell_1$ norm of the **substochastic** SR (SSR) could approximate the reciprocal of the visitation number, where SSR incorporates an additional “phantom” transition from each state, i.e., dropping the $\frac{1}{n(s)+1}$ probability in the transition function and making it underestimate the real SR. Only when $n(s) \rightarrow \infty$, SSR could be regarded as a proxy of SR. However, just in this case, the visitation number term (i.e., $\frac{\gamma}{n(s)+1}$) in the bound of SSR's $\ell_1$ norm will also gradually diminish to 0. Moreover, when $n(s)$ is small, the $\ell_1$ norm of SSR can capture the information of the visitation number but the $\ell_1$ norm of SR would not.
> >
> > We have modified the reference to the SR method as suggested by the reviewer.
> >
> > **Q10.** How was Figure 6 generated?
> >
> > **A10.** The trajectories in Figure 6 were collected by the policy learned by our approach after 5M steps, and the upper and lower panels used the same trajectories. We utilized those trajectories to evaluate the representations learned at different timesteps.
> >
> > **Q11.** *In the ablations, were the parameters of the ablated methods tuned?*
> >
> > **A11.** Yes, the ranges for tuning the hyper-parameters are listed in Appendix B.
> >
> >
> > ### Reference
> >
> > [1] Machado et al. "Count-Based Exploration with the Successor Representation" in AAAI 2020.

---

> > > ### Comment · Reviewer_EgA9 · 2021-11-20
> > > **Confidence interval**
> > >
> > > Thank you for going over my comments. I don't think my comments about the confidence interval were answered though. The function used was referenced, but what was the method used? Was it bootstrapping? The concern is about the low number of samples.

---

> > > > ### Author Response · Authors · 2021-11-20
> > > > **Clarification about confidence interval**
> > > >
> > > > Thank you for the reply!
> > > >
> > > > Yes, the confidence intervals in this work are bootstrap confidence intervals. We compute the bootstrap confidence intervals over five random seeds, which has been commonly applied in a number of previous works [1,2,3,4].
> > > > Specifically, we create $1000$ bootstrap resamples, and each resample is the mean of five samples drawn with replacement. Then we calculate the $(50-95/2, 50+95/2)$ percentile of these bootstrap resamples as the 95% confidence interval.
> > > >
> > > > If the reviewer has any additional questions, please let us know, and we would be happy to have further discussions.
> > > >
> > > >
> > > >
> > > > ### Reference
> > > >
> > > > [1] Lee et al. "Weakly-Supervised Reinforcement Learning for Controllable Behavior." in NeurIPS 2020.
> > > >
> > > > [2] Khadka et al. "Evolutionary Reinforcement Learning for Sample-Efficient Multiagent Coordination." in ICML 2020.
> > > >
> > > > [3] Li et al. "Hierarchical Reinforcement Learning with Advantage-Based Auxiliary Rewards." in NeurIPS 2019.
> > > >
> > > > [4] Mazoure et al. "Leveraging Exploration in Off-Policy Algorithms via Normalizing Flow." in CoRL 2019.

---

> ### Comment · Reviewer_EgA9 · 2021-11-29
> **Final Assessment**
>
> I've gone over the authors' rebuttal. I acknowledge them and I confirm this is my final score/recommendation for this paper.

---

### Official Review · Reviewer_GV2u · 2021-11-02

**Correctness:** 4
**Technical Novelty And Significance:** 3
**Empirical Novelty And Significance:** 3
**Recommendation:** 6
**Confidence:** 3

**Main Review:**

**Pros**
1. Comparisons between the proposed method and other hierarchical methods demonstrate that the algorithm results in better performance.
2. The authors performed thorough ablations demonstrating the impact of each proposed component of their algorithm.

**Cons**
1. The authors do not explain how the counts and potential measures are estimated from data. In particular:
    1. How are the cumulative counts $N(s)$ in (4) estimated, given $\pi_{hier}$ is changing over the course of training?
    2. How is $U(g_t)$ estimated from buffer data, given that the expectation is calculated with $g_e$ being set as a subgoal for the policy, and thus would not have been observed in the actual environment rollouts?
2. Why is prioritized sampling used in Equation 3? The motivation on this point was not really explained in detail.
3. For the ablative analysis, it seems like it would be better to evaluate reactive exploration using cumulative counts instead of immediate counts to better isolate the impact of reactive exploration versus learning a policy to maximize the same intrinsic rewards.

**Clarification Questions**
1. Why does choosing $\lambda(s)$ as a continuous function of the representation loss impose heavy computational demands? It seems like the losses are already being computed in the process of obtaining the triplets with minimal representation losses.
2. How is the latent space partitioned into cells if there are no knowns bounds on $\lVert \phi(s) \rVert$ a priori?
3. In motivating the potential measure, the authors claimed that the “novelty measure is a mixture of counts in the past and current representation spaces”, but it is unclear why this is the case if one can easily recompute $n$ when $\phi$ changes.
4. How is the low-level policy training done? Is hindsight experience replay used?

**Summary Of The Paper:**

The authors propose a hierarchical RL algorithm which augments an existing contrastive learning-based subgoal representation objective with heuristics for exploration. The proposed algorithm seeks to reduce representation drift over the course of learning by penalizing the learner for modifying $\phi(s)$ for states $s$ with low contrastive loss. Furthermore, the authors propose exploration heuristics that encourages the learner to explore in promising areas of latent space by combining count-based novelty and potential measures. The proposed algorithm is demonstrated to have the desirable properties, and outperforms existing methods. The analysis is complemented by an ablative analysis that disentangles the effects of each proposed mechanism.

**Summary Of The Review:**

Overall, I vote for a weak accept. The ideas in the paper are interesting, and the experimental evaluation is thorough and demonstrates the benefits of the proposed algorithm. However, the work could benefit from a more detailed description of how the relevant measures are estimated, as well minor changes to the experimental procedure.

---

> ### Author Response · Authors · 2021-11-17
> **Response to Reviewer GV2u**
>
> Thank you for the thoughtful review and positive feedback.
>
> **Q1.**
> **Q1.1.** How are the cumulative counts $N(s)$ in Equation (4) estimated, given $\pi_{hier}$ is changing over the course of training?
>
> **A1.1.** $N(s)$ is estimated with data from the replay buffer. As $\pi_{hier}$ is changing, we decrease the effects of the older rollouts in the replay buffer with a discount factor, taking inspiration from the Exponential Moving Average (EMA) method. We have added this implementation detail in Appendix B.
>
> An alternative way to deal with this off-policyness is to adopt the Importance Sampling (IS) technique, i.e., $N(\phi(s_i)) = E_{\mathcal{B}}[\sum_{j=0}^{\lfloor (T-i)/c \rfloor}\gamma^j \rho^{is} n(\phi(s_{i+jc}))]$, where $\rho^{is} = \prod_{t=0}^{(i+jc)/c} \pi^h_{new}(g|s_{tc}) / \pi^h_{old}(g|s_{tc})$, $\pi^h_{new}$ is the current high-level policy, $\pi^h_{old}$ is the old high-level sampling policy at state $s_{tc}$, and $\mathcal B$ is the replay buffer.
>
> In Appendix E.3, we compare the learning performance using the EMA estimation with the IS estimation in the Ant Maze task, and find that those two methods empirically demonstrate comparable performance.
>
> **Q1.2.** How is $U(g_t)$ estimated from buffer data, given that the expectation is calculated with $g_e$ being set as a subgoal, and thus would not have been observed in the actual rollouts?
>
> **A1.2.** Although we do not have the low-level rollouts exactly conditioned on $g_e$, as we have discretized the subgoal space to cells, the agent could have already collected rollouts conditioned on the subgoals in the same cell of $g_e$. So in practice, we use those rollouts to estimate $U(g_t)$.
> If there is no previous subgoal in the cell of $g_e$, we give $U(g_t)$ a pessimistic estimation by setting it to a small value.
>
> **Q2.** Why is prioritized sampling used in Equation 3?
>
> **A2.** Since we are dealing with hard exploration problems with a single start state, the state distribution in the replay buffer is imbalanced. To improve representation learning efficiency on this imbalanced buffer, we adopt the prioritized sampling technique.
>
> **Q3.** Evaluate reactive exploration using cumulative counts instead of immediate counts.
>
> **A3.** We have modified this ablation analysis in Section 5.4, and compare to reactive exploration using cumulative counts. Our approach still outperforms the modified reactive exploration method.
>
> **Clarification Questions**
>
> **Q4.** Why does choosing $\lambda(s)$ as a continuous function of the representation loss impose heavy computational demands?
>
> **A4.** We have modified this narrative in Section 3.1. By making $\lambda(s)$ a binary function, we could sharply distinguish the well-optimized embeddings from the underfitting ones. In Appendix E.4, we have added an empirical comparison between the binary $\lambda(s)$ and continuous $\lambda(s)$: $\lambda(s_t)=\lambda_0 / \sqrt{1 + L_{tri}(s_t, \cdot, \cdot)}$, and their learning curves are similar.
>
> **Q5.** How is the latent space partitioned into cells if there are no known bounds on $||\phi(s)||$?
>
> **A5.** We discretize the learned low-dimensional representation space with a mesh of a fixed density, and then those discrete embeddings are projected into a finite hash table, which does not require boundaries on $||\phi(s)||$.
>
> **Q6.** The authors claimed that "the novelty measure is a mixture of counts in the past and current representation spaces", but it is unclear why this is the case if one can easily recompute $n$ when $\phi$ changes?
>
> **A6.** $n(s)$ is estimated with the data in the buffer. Recomputing $n$ is **not** scalable, since the buffer size in deep reinforcement learning is usually large, and inferencing the representation network also costs time especially when the observation dimension is high.
>
>
> **Q7.** How is the low-level policy training done? Is hindsight experience replay (HER) used?
>
> **A7.** The low-level policy is learned with the Soft Actor-Critic method, as stated in Section 5.
>
> We do not use HER for training low-level policies, as the low-level rewards in our approach are dense. The HER paper has not reported a superior performance in dense reward settings.

---

### Official Review · Reviewer_yhF9 · 2021-11-02

**Correctness:** 4
**Technical Novelty And Significance:** 3
**Empirical Novelty And Significance:** 3
**Recommendation:** 8
**Confidence:** 4

**Main Review:**

1.

I think this paper is interesting and explores a novel set of ideas. The baselines also seem reasonable. The closest baseline in terms of using directional goal vectors is Feudal Networks. I would have expected to see a head to head comparison with this approach, even though this proposed method goes beyond it. However, the core idea of having a meta controller output goal vectors and then sub-controllers learning to execute them was explored in Feudal networks.

2.

What are the effects of changing the option termination condition? Currently it is hard coded to be c. What are the implications of this? Do the authors observe any deviations or improvements if this hyper parameter varies. It seems like the potential function, novelty measure and option termination are deeply interlinked. It would have been good to more clearly understand the relationship between these measures.

3.

Figure 4 is the main quantitative figure. It seems important to test the effects of stability regularization. This is highlighted qualitatively in Figure 6 but not shown in Figure 4.

4.

The qualitative analysis on the effects of the interaction between potential and novelty measure is quite sparse. It is not clear how it fails and where it works. Figure 6 is helpful but it needs improvements in terms of clarity and scope (other environments)

5.

Figure 5 is truncated at 5 million steps. How does the asymptotic performance look like for this method? Does it plateau sooner than baselines? What is the maximum achievable reward for these tasks?

**Summary Of The Paper:**

1. This paper investigates learning stable subgoals within a deep hierarchical reinforcement learning setup.

2. Two controllers are learned from the same experience replay buffer. The high level controller serves as a meta controller and the low level controller serves as a goal-achieving agent. The high level controller communicates abstract goals to the low level controllers.

3. The high level controller is optimized using an extrinsically specified reward function. The low level controller optimizes the intrinsic goal communicated by the high level controller. The subgoals are changed after a deterministic time length (known option termination).

4. The subgoals are designed with the key insight that "desirable novel subgoals should be reachable and effectively guide
the agent to unexplored areas.". Typically count-based, predicted or successor feature based rewards have been used as novelty measures. However, these fall short in terms of reasoning about reachability of states. To handle this, a potential measure for subgoals is proposed which regularizes the novelty measure.

5. To go to unexplored states, a directional goal is synthesized/imagined using the current state and a directional vector. The potential function makes sure that this is approximately reachable by formulating reward as the expected negative distance between the ending state and imagined goal. This is similar to Feudal networks (Vezhnevets et al.) but goes beyond it to handle diversity and reachability.

6. The approach is validated on a set of hard to explore continuous environments with reasonably strong and relevant baselines.

**Summary Of The Review:**

This paper presents an interesting and novel idea at the intersection of deep HRL, novelty based exploration and reachability. The experiments are sound but could require further clarification and expansion of scope. The clarity of the paper can also be improved to more directly address the need and important of stability regularization.

---

> ### Author Response · Authors · 2021-11-17
> **Response to Reviewer yhF9**
>
> Thank you for the constructive comments.
>
> **Q1.** A head-to-head comparison with the FeUdal Networks (FuN) approach.
>
> **A1.** We have run the FuN baseline in the Ant Maze task, and its convergent success rate after 70 million steps is zero. Similar experimental results were also reported in a previous HRL work [1]. In FuN, the subgoal representation learning does not encode extra inductive bias (e.g., the temporal coherence bias), and thus the low-level intrinsic reward (i.e., the distance in the latent subgoal space) is not consistent with the transition number in MDP. We plot the subgoal representation learned by FuN in Appendix D. We observe that the subgoal representation learned by FuN is noisy and could hardly provide proper supervision to low-level policy learning.
>
> **Q2.** What are the effects of changing the option termination condition? What are the implications of hard coding it to be $c$? Any deviations or improvements if this hyperparameter varies?
>
> **A2.** Most GCHRL works [1, 2, 3] use a fixed low-level policy length $c$, which helps avoid the burden of learning the termination function in options.
>
> As discussed in Section 5.5, $c$ trades off the complexity of training high-level and low-level policies. The ablation study in Figure 9(d) demonstrates that it is easy to find a workable range for $c$, and when $c$ is in a relatively large range, the performance may vary.
>
>
> **Q3.** Test the effects of the stability regularization quantitatively.
>
> **A3.** In Section 5.4, we have ablated the effects of the stability regularization by removing it. From the learning curves in Figure 7, we can see that the stability regularization has more significant effects in the tasks with image inputs, since image-version environments have induced more difficulty in representation learning.
>
> **Q4.** The qualitative analysis on the effects of the interaction between potential and novelty measures is sparse. It is not clear how it fails and where it works.
>
> **A4.** We have added a qualitative analysis on potential and novelty measures in the Ant Push task in Appendix E.7. From the visualization of subgoal selection w/ and w/o the potential measure, we find that the potential measure serves as a regularizer to the novelty measure, making the agent avoid exploring novel but unreachable states. For example, as shown in Figure 21(a), the subgoals selected w/o the potential measure, i.e., those selected only using novelty, may be close to the walls because the visitation numbers in these areas are less. Figure 21(b) shows that these less reachable and unpromising areas near the walls have lower potential, and the combination of those two measures effectively guides the agent to unexplored reachable areas, as shown in Figure 21(c).
>
> **Q5.** Figure 5 is truncated at 5 million steps. How does the asymptotic performance look like for this method? Does it plateau sooner than baselines? What is the maximum achievable reward for these tasks?
>
> **A5.** Does the comment mean that in **Figure 4**, the learning curves in the Ant Maze and Ant Push tasks are truncated at 5 million steps? Figure 5 is a visualization of the environment. If we misunderstood the comments, please let us know.
>
> To demonstrate the asymptotic performance of our method and the baseline methods, we update the results in Ant Maze, Ant Push, and their image versions to 10 million steps in Figure 4. The updated experimental results indicate that our approach converges significantly faster than the baselines, and HESS achieves better or comparable  asymptotic performance than all baselines, e.g., the intrinsically motivated exploration methods.
>
> The y-axis in Figure 4 represents the average success rate in 10 episodes, whose maximum value is $1.0$.
>
> ### Reference
>
> [1] Nachum et al. "Data-Efficient Hierarchical Reinforcement Learning." in NeurIPS 2018.
>
> [2] Nachum et al. "Near-Optimal Representation Learning for Hierarchical Reinforcement Learning." in ICLR 2019.
>
> [3] Li et al. "Learning Subgoal Representations with Slow Dynamics." in ICLR 2021.

---

> ### Comment · Reviewer_yhF9 · 2021-11-29
> **acknowledgements**
>
> I have gone over the rebuttal and am satisfied with the response -- my score is 8 (accept)

---

### Official Review · Reviewer_XmcD · 2021-11-03

**Correctness:** 3
**Technical Novelty And Significance:** 3
**Empirical Novelty And Significance:** 3
**Recommendation:** 6
**Confidence:** 4

**Main Review:**

Strengths:
- I appreciate that this paper studies the subgoal learning instability and high-level exploration which are of importance to GCHRL research.
- The regularization method for representation stability is reasonable, simple but empirically effective. Meanwhile, the representation instability is a problem encountered in many other scenarios and the proposed regularization method is general and of potentials to be leveraged in other representation learning problem.
- This paper proposes effective active exploration for high-level exploration of GCHRL. To my knowledge, the subgoal perturbation along with the definition of potential is new in GCHRL. I appreciate the combination of novelty and potential which properly takes novelty and reachability into consideration for an effective exploration selection.
- The experiments are extensive, well evaluating and demonstrating the characteristics of HESS across multiple perspectives.


&nbsp;


Weaknesses:
- I think the methods proposed in this paper are relatively simple and somewhat incremental, however, thanks to the solid experiments, the effectiveness of these methods are demonstrated. At a first glance, the representation regularization seems to be disconnected to the active exploration method. Later, I found that the stable representation learned is important to the effectiveness of novelty calculation. I recommend the authors to make the connection more obvious for a better convey of the story.
- Although I think the methods are reasonable in an overall view, I have a few concerns on concrete implementations. I list my questions and concerns below.


&nbsp;


My first concern is the calculation of novelty (Equation 4). I have no question on the maintenance of $n(\phi(s))$; but for the calculation of accumulated visit count of high-level state trajectory, I wonder given a state $s_i$, how the trajectory of policy $\pi_{hier}$ is obtained exactly?

&nbsp;

Second, for Equation 5, since the potential is defined over the expectation of high-level transition obtained by $\pi_{hier}$ with the perturbed subgoal $g_e$, how are such transitions obtained?

&nbsp;


For both above two concerns, one possible way is to simulate the rollouts with a world model, but this seems not to be the way used in this paper. Alternatively, are these approximated with the trajectories in the replay buffer? If so, how should consider the off-policyness and suboptimality?

&nbsp;

The third question is on the computation complexity. The top $k%$ selection in representation regularization and the calculation in Equation 6 (the selection of candidates according to the constraints, the calculation of novelty and potentials). It seems the computation is heavy for these. What are the practical implementations?

&nbsp;

Besides, I have a few questions on the experiments.
- How to understand that some baseline algorithms work better in image-version environments? E.g., H-SR/H-ICM on Ant-Maze and LESSON on Ant Push.
- Is the sentence ‘so the intrinsic rewards of H-ICM may vanish before the policies have been learned well’ checked in the experiments?
- In Figure 6, are the same 5 trajectories used for the upper and lower panels at each time point? And what are the trajectories exactly, since at the beginning of learning, the agent fails to reach the final goals (according to the results in Figure 4)?


&nbsp;


Minors:
- Can the authors explain more on the sentence ‘to keep the low-level reward function $r^l$ stationary while learning $\phi$, we concatenate states and state embeddings together as the low-level states’ (above Equation 1)?

&nbsp;


I will raise my score if my questions and concerns mentioned above can be well addressed.

================================

Post-rebuttal comments:

Some of my concerns and questions are well addressed. I raised my evaluation to a borderline acceptation although my main concern on the relatively complex mechanisms involved in the implementation, e.g., hash table, iterative sampling and fitering, table look-up and so on (a few of these are sample-wise), remains. And I think these computation implementation should be noted and described in detail in the revision later. However, I recognize the authors' efforts in pushing the boundary of HRL.


**Summary Of The Paper:**

The paper studies Goal-conditioned Hierarchical RL (GCHRL) and proposes a new algorithm called Hierarchical Exploration approach with Stable Subgoal representation learning (HESS) to improve the stability of subgoal representation learning and strengthen the exploration at high level. HESS is built on previous method LESSON. The instability of subgoal representation learning is alleviated by a representation regularization which is utilized to encourage the representation to be stable for the states with relatively lower triplet losses (originated from LESSON). Further, this paper proposes an active exploration method for the high-level learning. The method is built on the definitions of Novelty and Potential of states, which corresponds to accumulated visit counts of high-level state trajectory and a negative distance to the perturbed subgoals. Extensive experiments are conducted in a few MuJoCo environments with sparse reward, demonstrating the superiority of proposed algorithm and the effectiveness of different ingredients.

**Summary Of The Review:**

I vote for a borderline acceptation after discussing with the authors. I recognize the authors' efforts in pushing the boundary of HRL although I still have some concerns on the complexity of the proposed methods and the practical computation cost.

---

> ### Author Response · Authors · 2021-11-17
> **Response to Reviewer XmcD (Part 1)**
>
> Thank you for the thoughtful comments. We provide clarification to your questions and concerns as below. We appreciate if you have any further questions or comments.
>
> **Q1.** Make the connection between the representation regularization and the active exploration more obvious.
>
> **A1.** We have refined the introduction to explicitly describe this connection in our revision. The stability of the learned representation space is essential to the effectiveness of the novelty measure in active exploration. Furthermore, stable representation learning could alleviate the non-stationary issue in HRL, which also promotes effective hierarchical exploration and learning.
>
> **Q2.** How are the novelty and potential measures (Equations 4 and 5) calculated?
>
> **A2.** The novelty and potential measures are estimated with trajectories in the replay buffer. When calculating the expectation in those equations, to deal with the off-policyness of the data in the buffer, we decrease the effects of the older rollouts with a discount factor, taking inspiration from the Exponential Moving Average (EMA) method. We have added this implementation detail in Appendix B.
>
> An alternative way to deal with this off-policyness is to adopt the Importance Sampling (IS) technique, i.e., $ N(\phi(s_i)) = {E_{\mathcal B}}[\sum_{j=0}^{\lfloor (T-i)/c \rfloor} \gamma^j \rho^{is}n(\phi(s_{i+jc}))]$, where $\rho^{is} = \prod_{t=0}^{(i+jc)/c} \pi^h_{new}(g|s_{tc}) / \pi^h_{old}(g|s_{tc})$, $\pi^h_{new}$ is the current high-level policy, $\pi^h_{old}$ is the old high-level sampling policy at state $s_{tc}$, and $ \mathcal B$ is the replay buffer.
>
> In Appendix E.3, we compare the learning performance using the EMA estimation with the IS estimation in the Ant Maze task, and find that those two methods empirically demonstrate comparable performance.
>
>
> **Q3.** The computation complexity of the top $k$ selection in the representation regularization and the calculation in Equation 6.
>
> **A3.** Practically we have made the following approximations to relieve the computation complexity.
>
> * Top $k$ selection:
>
>   To reduce the computation complexity, we conduct the top $k$ selection every 100 episodes. During the interval, we approximate it with the previous selection. The top $k$ selection sharply distinguishes the well-optimized embeddings from the underfitting ones.
>
>   We have also tried an alternative soft implementation by weighting $\lambda(s)$ continuously based on the representation loss, i.e., $\lambda(s_t)=\lambda_0 / \sqrt{1 + L_{tri}(s_t, \cdot, \cdot)}$. In this soft implementation, we do not need to make the top $k$ selection anymore. We compare these two implementations in Appendix E.4, and they show similar learning performances.
>
> * The calculation in Eq. 6:
>
>   **The selection of candidates according to the constraints**:
>
>   In practice, we do **not** take the $argmin$ operator upon all the states in the buffer satisfying the constraint. Instead, we just sampled a batch of states from the replay buffer. The detail is described in Appendix B.
>
>   **The calculation of the novelty and potential measures**:
>
>   We update the estimation of the novelty and potential measures every 10 episodes to alleviate their computation.
>
> **Questions on the experiments:**
>
> **Q4.** How to understand that some baseline algorithms work better in image-version environments?
>
> **A4.** The image observations with relatively high dimensions have induced difficulty in representation learning. However, since walls and movable objects around the agent are included in the top-down images (which are not included in the vector-based settings), the policy learned with such image inputs may generalize better to some extent. Furthermore, in the image-version tasks, the observations are preprocessed to low-resolution images, similar to the strategy taken in previous works [1, 2].
>
> In Appendix E.5, we compare the generalization ability with image inputs and non-image inputs by setting the start state in a less-explored area. We find that the policy with image inputs avoids the walls better than that with non-image inputs.
>
> **Q5.** Is the sentence "so the intrinsic rewards of H-ICM may vanish before the policies have been learned well" checked in the experiments?
>
> **A5.** We have added a visualization of the H-ICM intrinsic rewards in Appendix D. At 0.1M and 0.3M timesteps, we sample 100 states from the replay buffer, and show the intrinsic rewards at those states. From Figure 11, we can see that at the early learning stage (0.3M steps), the H-ICM intrinsic rewards at most states become small, which is largely due to the fast learning of the dynamic model.
>
> ### Reference
>
> [1] Nachum et al. "Near-Optimal Representation Learning for Hierarchical Reinforcement Learning." in ICLR 2019.
>
> [2] Li et al. "Learning Subgoal Representations with Slow Dynamics." in ICLR 2021.

---

> > ### Author Response · Authors · 2021-11-17
> > **Response to Reviewer XmcD (Part 2)**
> >
> > **Q6.** In Figure 6, are the same 5 trajectories used for the upper and lower panels at each time point? And what are the trajectories exactly?
> >
> > **A6.** Yes, the upper and lower panels use the same trajectories. Those trajectories are collected by the policy learned by our approach after 5M steps, and they are only used to evaluate the representation learning.
> >
> > **Minors:**
> >
> > **Q7.** Explain more on the sentence 'to keep the low-level reward function $r^l$ stationary while learning $\phi$, we concatenate states and state embeddings together as the low-level states'?
> >
> > **A7.** There was a typo in the previous version. The low-level reward actually is $r^l(s_t, \phi(s_{t}), g_t) = -D(g_t, \phi(s_{t}))$, which is changing since the representation $\phi$ is updated online. To satisfy the Markov property of the reward function, we add $\phi(s)$ to the input of the low-level policy.
> > In Appendix E.6, we compare the learning performance of the two kinds of low-level state space ($s$ v.s. $s||\phi(s)$) in the Ant Push (Images) task. Including $\phi(s)$ in the low-level state space leads to slightly faster convergence.

---

> > ### Comment · Reviewer_XmcD · 2021-11-18
> > **Further Discussions on Novelty/Potential Calculation and Subgoal Selection**
> >
> > I appreciate the authors' detailed response and additiona experiments in the revision, which does a great favor in addressing some of my concerns.
> >
> > After reading the response, I want to discuss more about the calculation of novelty and potential, and the subgoal selection.
> >
> > 1. The calculation of novelty and potential
> >
> > For the first way, i.e., to estimate with trajectories in the replay buffer, I think we need to find the trajectories which start with similar (since exact equivalence is not possible for continuous state) states to the state $s_t$ we are going to estimate. I wonder how this is exactly done.
> >
> > For the second way, i.e., trajectory IS, I think it may not be adequate to only involve the IS ratio of the high-level policy $\pi^{h}$ since the trajectories are generated by historical high- and low-level policies. Moreover, I think calculating the trajectory-to-go IS is also not sufficient since the occupancy distribution of the state $s_t$ considered is different for current hierarchical policy and the old ones [1].
> > Similar concerns are also raised for the calculation of potential in Eq.5 while only c-step trajectories are involved.
> >
> > I want to know what the opinions of the authors on these are.
> >
> >
> > 2. The subgoal selection
> >
> > As provided in Appendix B, the state candidates are obtained by firstly sampling $M=1000$ states satisfying the constraint from the buffer. First, I wonder how this is done. Since we do not calculate all the states in the buffer, it seems that we need to iteratively do batch sampling-filtering-batch-sampling-filtering-... until we filter out $M$ states  satisfying the constraint. Do I understand it right? If so, how many iterations or what is the total number of the sampled states?
> >
> > Another concern is that, why is $M=1000$ sufficient to ensure a good performance of such subgoal selection mechanism, since the buffer has a size of $1e6$?
> >
> > Moreover, for subgoal selection, the estimation of the novelty and potential measures are updated every 10 episodes to alleviate their computation. My question is that, since the meansures are state-dependent and the $M=1000$ candidates are to be sampled for each subgoal selection step, does such an alleviation means that we need to keep a dictionary (or some memory in other form) for each possible state to store the estimated novelty and potential measures within 10 episodes?
> >
> > &nbsp;
> >
> > I am looking forward to further discussion.
> >
> > &nbsp;
> >
> > Reference:
> >
> > [1] Richard S. Sutton, Ashique Rupam Mahmood, Martha White. An Emphatic Approach to the Problem of Off-policy Temporal-Difference Learning. J. Mach. Learn. Res. 17: 73:1-73:29 (2016)

---

> > > ### Author Response · Authors · 2021-11-20
> > > **Further Clarifications**
> > >
> > > Thank you for the thoughtful and prompt reply! We provide further clarifications as below.
> > >
> > > **Q1. The calculation of novelty and potential**
> > >
> > > **Q1.1** How to find trajectories which start with similar states to the state $s_t$ we are going to estimate?
> > >
> > > **A1.1** To estimate novelty, our method utilizes cell discretization in the learned representation space, as stated in Section 3.2. The states in the same cell are regarded as similar, and trajectories starting from those similar states are used to calculate the expectation.
> > >
> > > **Q1.2 Trajectory IS**
> > >
> > > **(1)** The IS ratio of the low-level policy is not involved.
> > >
> > > We have run the experiment with the IS implementation including the low-level IS ratio. The performance of including the low-level IS ratio is not as good as the previous ones, as shown in Figure 17, since the importance weight suffers from the *curse of horizon*, which is discussed in Section 2 of [1].
> > >
> > > **(2)** The occupancy distribution is different for the current hierarchical policy and the old ones.
> > >
> > > The mismatch of the state distribution does not affect the calculation of the trajectory IS, since we compute the expectation on trajectories, but not states, i.e.,
> > > $$N_{\pi_{new}}(s_t) = E_{\tau\sim \pi_{new}}[\sum_{j=0}^{\lfloor (T-t)/c \rfloor} \gamma^j n(\phi(s_{t+jc}))]$$
> > > $$ = E_{\tau\sim \pi_{old}} [\frac{p_{\pi_{new}}(\tau|s_t)}{p_{\pi_{old}}(\tau|s_t)}\sum_{j=0}^{\lfloor (T-t)/c \rfloor} \gamma^j n(\phi(s_{t+jc}))] $$
> > >  $$  = E_{\tau \sim \mathcal \pi_{old}}[\sum_{j=0}^{\lfloor (T-t)/c \rfloor} \gamma^j \rho_{t+jc}(\tau) n(\phi(s_{t+jc}))],$$
> > >
> > > where $\tau = (s_t, a_t, s_{t+1}, \dots)$ is the state-action trajectory from the start state $s_t$, $p(\tau|s)$ denotes the trajectory distribution given the start state, and $\rho_{t+jc}(\tau)$ is the importance weight for the hierarchical policy,
> > > $$
> > > \begin{aligned}
> > > \rho_{t+jc}(\tau) &= \prod_{i=0}^{t+jc} \pi_{new}(a_i|s_i) / \pi_{old}(a_i|s_i)
> > > &=\prod_{i=0}^{\lfloor(t+jc)/c\rfloor -1} \frac{\pi^h_{new}(g|s_{ic})\prod_{m=ic}^{(i+1)c}\pi^l_{new}(a|s_m, g)}{ \pi^h_{old}(g|s_{ic})\prod_{m=ic}^{(i+1)c}\pi^l_{old}(a|s_m, g)}.
> > > \end{aligned}
> > > $$
> > >
> > > In contrast, if the expectation is taken over states, the ratio between the new state distribution and the old one needs to be considered:
> > >
> > > $$ N_{\pi_{new}}(s_t)  = E_{\tau\sim \pi_{new}}[\sum_{j=0}^{\lfloor (T-t)/c \rfloor} \gamma^j n(\phi(s_{t+jc}))]$$
> > > $$ = E_{s^h \sim d^h_{new}}[n(\phi(s^h))]\quad (when~ T\rightarrow\infty) $$
> > > $$  = E_{s^h\sim d^h_{old}}[ \frac{d^h_{\pi_{new}}(s^h)}{d^h_{\pi_{old}}(s^h)} n(\phi(s^h))],$$
> > >
> > > where $s^h$ is the high-level state and
> > > $d^h_{\pi}(s^h)$ is the *stationary state distribution* in the high level under policy $\pi$ from the start state $s^h$. This derivation is similar to Eq. (4) in [1], which encodes the discount factor $\gamma$ into the stationary distribution.
> > >
> > > Eq. (3) and Eq. (6) in [1] also demonstrate this difference when $n(\phi(s))$ is replaced by $r(s, a)$.
> > >
> > >
> > > **Q2. The subgoal selection**
> > >
> > > **Q2.1** We need to iteratively do batch-sampling-filtering-... Do I understand it right? What is the total number of sampled states?
> > >
> > > **A2.1** Yes, this understanding is right. The total number of sampled states varies when the subgoal selection is conducted at different states. Empirically we find that the total sampled number at most states is around $2000$, and when the visitation density around the subgoal selection state is low, the total number could be as large as $7000$.
> > >
> > >
> > > **Q2.2** Why is $M=1000$ sufficient to ensure a good performance, since the buffer has a size of $1e6$?
> > >
> > > **A2.2** As the subgoal selection is restricted to the neighborhood of the current latent state, how large $M$ is dependent on the radius $r_g$ of the subgoal selection neighborhood, but not closely relevant to the buffer size.
> > >
> > > When sampling candidate goals from the replay buffer, one related work in multi-goal RL [2] only uses a size of $100$ for the candidate set in their experiments. $M$ in our work is much larger than $100$, which would be sufficient.
> > >
> > > **Q2.3** Do we need to keep a dictionary for each possible state to store the estimated novelty and potential?
> > >
> > > **A2.3** We store the estimated novelty and potential in a hash table, and update it every $10$ episodes. During the update interval, the subgoals are selected using the previous estimation. The newly collected trajectories are stored in the replay buffer, and then used to update the novelty and potential periodically.
> > >
> > >
> > > If the reviewer has any additional questions, please let us know and we would be happy to have further discussions.
> > >
> > > ### Reference
> > >
> > > [1] Liu et al. "Breaking the Curse of Horizon:
> > > Infinite-Horizon Off-Policy Estimation." in NeurIPS 2018.
> > >
> > > [2] Pitis et al. "Maximum Entropy Gain Exploration for Long Horizon Multi-goal Reinforcement Learning." in ICML 2020.

---

> > > > ### Comment · Reviewer_XmcD · 2021-11-21
> > > > **Response to Authors' Response on Further Clarifications**
> > > >
> > > > I appreciate the authors' response. Most of my questions are addressed and confirmed.
> > > >
> > > > I think my main concern is on the relatively complex mechanisms involved in the implementation, e.g., hash table, iterative sampling and fitering, table look-up and so on (a few of these are sample-wise). However, I recognize the non-triviality in pushing the boundary of HRL and I appreciate the authors' efforts. Therefore, I will consider to raise my rating.

---

> > > > > ### Author Response · Authors · 2021-11-22
> > > > > **Thank you for appreciating the contribution of this work!**
> > > > >
> > > > > Thank you for considering to raise the rating! We really appreciate your efforts in reviewing this paper and the thoughtful comments that are helpful for improving our work.

---

### Decision · Program_Chairs · 2022-01-20

**Decision:**

Accept (Poster)

**Comment:**

The paper proposes a new goal-conditioned hierarchical RL method aimed at improving performance on sparse reward tasks. Compared to prior work the novelty lies in a new way of improving the stability of goal representation learning and in an improved exploration strategy for proposing goals while taking reachability into account.

The paper does a good job of motivating the main ideas around stability and combining novelty with reachability. Reviewers found the quantitative evaluation and the choice of baselines to be good with the exception of not including Feudal Networks which the authors explained was due to poor performance on the hard exploration tasks (something that has been observed in prior work). Reviewers also found the thoroughness of the ablations and insightful visualizations to be highlights. Overall, reviewers were unanimous in recommending acceptance, which I support.